# Physicochemical, Pasting Properties and In Vitro Starch Digestion of Chinese Yam Flours as Affected by Microwave Freeze-Drying

**DOI:** 10.3390/foods11152324

**Published:** 2022-08-03

**Authors:** Linlin Li, Junliang Chen, Danqi Bai, Mengshuo Xu, Weiwei Cao, Guangyue Ren, Aiqing Ren, Xu Duan

**Affiliations:** 1College of Food and Bioengineering, Henan University of Science and Technology, Luoyang 471023, China; linlinli2020@126.com (L.L.); junliangchen@126.com (J.C.); baidanqi0121@163.com (D.B.); xumengshuo2022@163.com (M.X.); caoweiwei@haust.edu.cn (W.C.); guangyueyao@163.com (G.R.); 2Institute of Food Research, Hezhou University, Hezhou 542899, China

**Keywords:** Chinese yam flour, microwave freeze-drying, physicochemical properties, starch digestibility, pasting properties

## Abstract

Microwave freeze-drying (MFD) is a new freeze-drying technique, which differs from single microwave treatment; it involves simultaneous effects of microwave power, time, and the moisture state applied to the materials. In this study, the effects of MFD under various microwave power densities (0.5, 1.0, and 1.5 W/g) on the drying characteristics of Chinese yam slices and the physicochemical, pasting, and thermal properties as well as the starch digestibility of the flour were investigated using conventional hot air drying (HAD) at 50 °C as a control. Compared to HAD, MFD shortened the drying time up to 14.29~35.71%, with a higher drying efficiency at a high microwave power density (1.5 W/g). MFD yam flours provided benefits over HAD products in terms of color, water/oil absorption capacity, and solubility, exhibiting high hot-paste viscosity but low resistant starch content. The content of total starch and free glucose of the yam flour and its iodine blue value were significantly influenced by the drying method and the MFD process parameters (*p* < 0.05). MFD processing could disrupt the short-range ordered structure of yam starch. Among the MFD flours, samples dried by MFD at 1.5 W/g presented the highest ratio of peak intensity at 1047 and 1022 cm^−1^ (R_1047/1022_) value, gelatinization enthalpy, and resistant starch content. These results gave a theoretical foundation for the novel freeze-drying method that MFD applied to foods with a high starch content, enabling the production of a product with the desired quality.

## 1. Introduction

Chinese yam (*Dioscorea opposita* Thunb.) is the tuber of *Dioscorea opposita*, a medicine–food homology species with excellent antibacterial, anti-inflammatory, and immunomodulatory properties [1]. It has attracted considerable attention because of its high nutritional and medicinal value. After harvesting, Chinese yams are mostly consumed in fresh form. However, they are susceptible to spoilage due to the high moisture content. To extend the shelf life, yams are also processed into canned and ready-to-eat snack foods. Drying is a technique to effectively alleviate the loss of fresh yams caused by spoilage and deterioration.

Hot air drying (HAD) is a common method for large-scale processing in the food industry. However, the technical defects of long drying time and substantial quality loss for HAD are prominent [2]. Freeze-drying (FD) is considered to be one of the best drying methods in terms of preserving the quality of the products. However, the expensive running costs limit its application in the food industry [3]. Microwave freeze-drying (MFD) is a novel freeze-drying developed on the basis of traditional FD, using microwaves as a heating source to provide heat for ice sublimation. Our previous studies have proven that MFD is effective in improving drying efficiency, reducing energy consumption, and maintaining good sensory and nutritional qualities of foods compared to FD [4,5,6].

Dried yams can be consumed as a dish after cooking or further processed into powder form (such as flour), which can be added to various food formulations as a functional food additive. Starch is the main component of yams, with a content of up to approximately 80%. The starch-related quality of yam flour is highly dependent on the drying method, which plays a crucial role in food formulations. Ahmed and Al-Attar [7] reported the effect of freeze-drying and tray-drying on rheological and structural properties of chestnut flour doughs and stated that freeze-dried chestnut flour maintained the microstructure of the dough and had better viscoelastic and pasting properties than tray-dried chestnut flour. Our previous work investigated the physicochemical, functional, and rheological properties of yam flours dried by MFD, microwave vacuum-drying (MVD), HAD, and FD. MFD was found to be superior to FD in terms of pasting properties and low-temperature gelation properties [8]. However, less attention has been paid to the drying process parameters of MFD, limiting the potential application of this technology in the processing of starchy foods.

Starch digestibility has been widely researched at present. Foods with low starch digestibility are considered to be healthy for their hypoglycemic effect. Several studies focused on the in vitro digestibility of starchy foods as affected by drying. Chen et al. [9] reported that when compared to starch isolated from fresh yam, HAD resulted in reduced resistant starch (RS) content, with higher temperatures leading to lower RS content, while FD resulted in increased RS content. Yang et al. [10] found that FD caused a decrease in the slow-digesting starch (SDS) content and an increase in the rapidly digestible starch (RSD) and RS contents of chestnut flour compared to HAD. Guan et al. [11] compared the effects of microwave vacuum-drying (MVD), HAD, and FD on the starch digestibility of yam flours and showed that the lowest RS content was obtained by MVD-processed flour. They attributed the reason to the high microwave drying temperature (90–100 °C), which increased the degree of gelatinization and made the starch more susceptible to enzymatic attack. However, the methods by which MFD parameters affect the digestibility of yam starch need to be discussed.

Even though the effects of microwave treatment on starch or starch-based foods have been extensively studied, focusing on microwave intensity or treatment time [12,13,14], the effects of MFD are still worth investigating. Because MFD differs from single microwave treatment, it involves more complicated microwave action, including the simultaneous effects of power, time, and the moisture state applied to the materials. The aim of this study was to investigate the effects of MFD process parameters on the physicochemical, thermal, and pasting properties of dried yam flours. Furthermore, the morphology and in vitro starch digestibility of yam flours were also evaluated.

## 2. Materials and Methods

### 2.1. Materials

Fresh Chinese yams (*Dioscorea opposita* Thunb.) were purchased from the local farmers in Wenxian County, Jiaozuo City, Henan Province. They were stored in a refrigerator at 4 ± 1 °C for later use. Before drying, the yams were washed, peeled, and then cut into slices with a thickness of 3 ± 0.5 mm.

### 2.2. Drying Experiments

A microwave freeze-dryer designed by our research team and customized by Nanjing Yatai Microwave Technology Co. was employed to perform the MFD for Chinese yam, with the specific construction of the equipment as described by Liu et al. [5]. Before MFD, the yam slices were placed in the ultra-low temperature refrigerator (−80 °C) for more than 8 h. For each run, 200 g of samples were used. The MFD was conducted under different microwave power densities (0.5, 1, and 1.5 W/g) with fixed parameters including the pressure of 100 Pa and cold trap temperature below −40 °C. Drying was carried out until the equilibrium moisture content of the sample was reached. Sample temperatures measured by optic fiber sensors were recorded. As a control, hot air-drying at a temperature of 50 °C and an air velocity of 1 m/s was performed until the mass of samples no longer changed. The dried yam slices were first ground in a pulverizer and then sieved through a 60-mesh screen to obtain yam flour for further analysis.

### 2.3. Determination of Physicochemical Properties

#### 2.3.1. Color

The color of samples was measured by a colorimeter (Ci7 × 00, X·rite technology Co. Ltd., Grand Rapids, MI, USA) after calibration. Results were recorded as *L**, *a**, and *b**, which represent brightness, red-green chromaticity, and yellow-blue chromaticity of the samples, respectively. The whiteness index (*WI*) of the sample was calculated by the following Equation (1):(1)$$WI=100-\sqrt{{\left(100-L^{*}\right)}^{2}+a^{*2}+b^{*2}}$$

#### 2.3.2. Water- and Oil-Absorption Capacity

One gram of flour sample and 10 mL of distilled water/peanut oil were mixed by a vortex mixer (VORTEX-5, Kylin-Bell Lab Instruments Co., Ltd., Nantong, China) for 5 min. After centrifuging at 4000 r/min for 10 min at room temperature, the supernatant was poured out. The mass of water/oil adsorbed by the sample was calculated. Water-absorption capacity (WAC) and oil-absorption capacity (OAC) were expressed as the weight of water/oil absorbed per gram of dry matter, respectively.

#### 2.3.3. Solubility

Distilled water (20 mL) was transferred to about 2.0 g of yam flour in a 100 mL beaker and stirred for 0.5 h at 50 °C using a magnetic heated stirrer. The resultant mixture was centrifuged at 3000 r/min for 10 min at room temperature. Then, the dry matter in 2 mL supernatant was measured. Solubility (*S*) was expressed as the weight of dissolved solids in supernatant per gram of dry matter.

#### 2.3.4. Bulk Density

About 5.0 g of yam flour was accurately weighed and poured into a 10 mL graduated cylinder. After gentle shaking until the flour stopped falling, the volume was recorded. The bulk density (*BD*) was calculated according to the following equation:(2)$$BD=\frac{m}{V}$$
where *m* is the sample mass (g); *V* is the volume of the sample (mL).

#### 2.3.5. Iodine Blue Value (IBV)

The determination of IBV was based on the method reported by Chen et al. [15] with some modifications. About 0.25 g of yam flour was dispersed in 50 mL distilled water. The mixture was heated to 65.5 °C and kept for 1 min, followed by centrifugation at 8000 r/min for 10 min at room temperature. Then, 1 mL of supernatant was gathered and transferred to a 10 mL colorimetric tube, followed by the addition of 1 mL iodine standard solution (0.02 mol/L) and then dilution with deionized water to the scale. IBV was expressed by the absorbance at 650 nm.

### 2.4. Pasting Properties

The pasting properties of yam flours were measured by a Brabender viscometer (803302, Brabender Technologie GmbH & Co. KG, Duisburg, German). Yam flour suspension with a moisture content of 6% (dry basis) was prepared for testing. The temperature program was set as follows: heating from 50 °C to 95 °C at a rate of 3 °C/min, followed by holding at 95 °C for 10 min, then cooling from 95 °C to 50 °C at a rate of 3 °C/min, and maintenance at 50 °C for 10 min. The pasting temperature (*T*_p_), peak time (t_p_), peak viscosity (PV), trough viscosity (TV), final viscosity (FV), breakdown viscosity (BD), and setback viscosity (SB) were obtained.

### 2.5. Thermal Properties

The yam dispersion was prepared by mixing flour and distilled water in a ratio of 1:3 (g/mL), which was sealed in an aluminum pan and equilibrated at room temperature for 24 h. The thermal properties were measured using a differential scanning calorimeter (DSC3, METTLE TOLEDO, Greifensee, Switzerland), as described in the previous study [8]. The gelatinization onset temperature (*T*_o_), peak temperature (*T*_p_), endpoint temperature (*T*_e_), and gelatinization enthalpy change (Δ*H*) were recorded.

### 2.6. Fourier Transform Infrared Spectroscopy (FTIR) Analysis

The samples were homogeneously mixed with KBr at a mass ratio of 1:100. Then, the tablets to be tested were made using a special mold. A Fourier transform infrared spectrometer (IS10, Nicolet, Madison, WI, USA) was employed to measure the spectrum in the range of 400~4000 cm^−1^ with KBr as background.

### 2.7. In Vitro Starch Digestibility Properties

In vitro starch digestion of yam flour cooked at 95 °C for 10 min was measured according to the method of Englyst et al. [16]. The percentage of RDS, SDS, and RS in total starch was calculated as follows:RSD = (G_20_ − FG) × 0.9/TS(3)
SDS = (G_120_ − G_20_) × 0.9/TS(4)
RS = [TS − (RDS + SDS)]/TS(5)
where G_20_ and G_120_ are the weight of hydrolyzed glucose after 20 and 120 min, respectively (mg). FG and TS are the weight of free glucose and total starch in samples (mg), respectively.

The total starch content was determined by enzymatic hydrolysis using total starch assay kit (Megazyme International Ltd., Bray, Ireland).

### 2.8. Scanning Electron Microscope (SEM)

A scanning electron microscope (TM3030Plus, Hitachi High-Tech Corporation, Tokyo, Japan) was used to measure the micromorphology of yam flour at a magnification of 1000× and 2000× and an accelerating voltage of 15 kV.

### 2.9. Statistical Analysis

All tests were repeated three times, and the results were expressed as mean ± standard deviation. One-way analysis of variance (ANOVA) and Duncan’s multiple range tests with a 95% confidence level (*p* < 0.05) were performed using SPSS 19.0. (SPSS Inc., Chicago, IL, USA) to evaluate the significant difference among the groups.

## 3. Results and Discussion

### 3.1. Drying Characteristics

The drying characteristics of yam slices by MFD under different microwave power density levels are shown in Figure 1. It took 360, 330, and 270 min for MFD at 0.5, 1.0, and 1.5 W/g, respectively. Drying was accelerated by the increasing microwave power density, thus shortening the drying time. Compared to HAD (420 min, data not shown), a higher drying efficiency was obtained by MFD although it was low-temperature sublimation drying. Rapid microwave heating provided energy for the sublimation of ice crystals [17]. As shown in Figure 1b, the material temperature increased continuously in the drying process, with a slow rising in the sublimation drying stage. Poor microwave absorption and conversion ability of materials caused by the low dielectric constant and loss factor at low temperature is the reason [18]. When the material temperature was greater than 0 °C, its ability to absorb microwaves was enhanced, which can be proved by the rapid temperature rising (Figure 1b). For MFD at different microwave power densities, the same change trend of material temperature was observed. The slower temperature rise was found in the case of the lower microwave power density, resulting in a longer drying time.

### 3.2. Physicochemical and Functional Properties

#### 3.2.1. Color and Bulk Density

Color is the intuitive perception of quality that food brings to the consumer and influences consumer preference and acceptance. The effects of different drying conditions on the color of dried yam flour were evaluated as shown in Table 1. There was a significant difference in color (including *L**, *a**, *b**, and *WI* values) of yam flour obtained by HAD and MFD (*p* < 0.05), while microwave power density had no significant effect on sample color (*p* ≥ 0.05). Compared to HAD samples, high *L*^*^ and *WI* values of the MFD samples were observed, indicating better brightness and whiteness. MFD operated at low temperatures and low-oxygen conditions, so the yams suffered less oxidation or/and enzymatic browning, which made MFD yams appear bright and white in color and more acceptable to consumers.

Bulk density is a complex character of powdered foods that is of great importance from the view of economy and functionality. It can be seen from Table 1 that there was a significant difference in bulk density for yam flour dried by HAD and MFD (*p* < 0.05). A high bulk density (0.67 g/mL) was observed in HAD-treated yam flour. The reason was that HAD caused the material to shrink seriously, resulting in the compact particle structure in the sample after pulverization, while MFD could better maintain the original shape of the material due to the sublimation drying [8]. However, no significant difference in bulk density among samples dried by MFD at different microwave power densities was found (*p* ≥ 0.05).

#### 3.2.2. Total Starch Content, Free Glucose Content, and IBV

Total starch content, free glucose content, and IBV were determined to estimate the effects of different drying methods on starch in yams (Figure 2). The total starch content of dried yam flours ranged from 67.74 to 74.86 g/100 g dry weight (DW), with the lowest value for MFD 1.5 W/g dried samples. On the contrary, the MFD 1.5 W/g dried samples had the highest free glucose content (14.26 g/100 g DW). More hydrolysis of starch to glucose as affected by microwave action may explain the reason. There was no significant difference in total starch content and free glucose content among MFD 0.5 W/g, MFD 1.0 W/g, and HAD dried samples (*p* ≥ 0.05). For HAD, a drying temperature of 50 °C favored the hydrolysis of starch to glucose by endogenous α-amylase and glucosidase in plants, as they had the highest enzyme activity at 55–60 °C [19]. However, for MFD, although the material temperature was lower, the starch cleavage was enhanced by microwave.

IBV is an indicator to evaluate the amylose content in starch-based foods because of its ability to complex with iodine. Results showed that the drying method had significant effects on the IBV of samples (Figure 2b). Higher IBVs of all the MFD samples than that of the HAD ones were observed, suggesting a higher amylose content in MFD samples. However, Chen et al. [19] found that FD reduced amylose content compared to HAD. Such different findings may be related to the various drying methods. The thermal effect of HAD led to more leaching of amylose, while FD operates at low temperatures, protecting amylose from high temperatures. However, the MFD samples with low-temperature drying had higher IBVs than HAD yams with high-temperature processing. The possible reason was that frictional forces generated by the rotation of polar molecules in the presence of microwave energy resulted in the breakage of the short-chain in amylopectin, producing more amylose [20]. In addition, there was a significant difference (*p* < 0.05) in the IBVs of yam samples treated with MFD at different microwave power levels, with the lowest IBV observed in samples dried by MFD at 1.0 W/g. This may be attributed to the longer drying time of MFD at 0.5 W/g and the higher microwave energy of MFD at 1.5 W/g, which resulted in more breakage of the short chain in amylopectin and increased amylose content.

#### 3.2.3. Functional Properties

The functional properties of yam flours were evaluated as presented in Table 2. Water absorption capacity measures the extent to which a powder can be incorporated into a food formulation. In general, powder samples with high WAC are expected in food processing applications. The WAC of MFD samples ranged from 3.31 to 3.50 g/g of dry matter, which were significantly higher than that of HAD samples (*p* < 0.05). The WAC is highly correlated with the integrity of the crystalline structure in starch, with lower crystallinity starches exhibiting a higher WAC [21]. According to Li et al. [8], the crystallinity of MFD-processed yams was lower than that of HAD samples. Sublimation of ice crystals in freeze-drying causes disruption of the double helix structure of starch molecules, which contributes to the decrease of relative crystallinity [22]. Likewise, when starch undergoes microwave irradiation, the internal molecular rotation induces the de-branching of branched starch, disrupting the crystalline structure [23]. Both of these effects are simultaneously involved in the MFD, thus leading to a reduction in crystallinity. However, no significant difference in WAC were found among three groups of MFD samples (*p* ≥ 0.05). It demonstrated that the microwave power level and duration of the MFD did not have significant effects on the WAC of the yam flour.

Since fats in food often act as flavor retainers and promote oxygen-mediated deterioration reactions, the oil-absorption capacity is important. As affected by drying methods, similar trends with WAC were found in OAC of yam flours. The lowest OAC obtained in the HAD sample was 0.24 ± 0.07 g/g of dry matter. OAC of the yam flours was affected by the structure of starch and protein. Starch does not contain polar groups, and its adsorption to oil is a physical capture depending on the shape and sizes of the starch [24]. The degree of denaturation of hydrophobic proteins affected by processing has a marked effect on OAC. The significant difference in OAC between HAD- and MFD-treated samples may be attributed to the different particle size and degree of protein denaturation. In contrast, microwave intensity of MFD had no significant effect on the OAC of yam flours (*p* ≥ 0.05).

Solubility is of great interest as an important characteristic in food processing. As shown in Table 2, the solubility of yam flours dried by HAD and MFD under three microwave power levels was significantly different (*p* < 0.05), with higher solubility in MFD-treated yam flours. When subject to microwave treatment, the damage of starch granules was enhanced, thus increasing the solubility of the starch [8]. Furthermore, as for the yam flour samples, the contribution of other components, such as protein and free glucose, to solubility cannot be ignored. In this study, the highest solubility found in the samples dried by MFD at 1.0 W/g was likely related to the better maintenance of soluble protein due to the suitable microwave intensity and drying time.

### 3.3. FTIR Spectral Analysis

The FTIR spectra provides the information of the main chemical components in the sample. The changes in crystal form, chain conformation, and helical structure of molecular starch will change the infrared absorption. Figure 3 shows the FTIR spectra of Chinese yam flours. All samples presented basically similar FTIR spectra. There was no new functional groups generated, but the position and intensity of the absorption peak were slightly different as affected by drying methods and processing parameters. The bands in the range of 1100–950 cm^−1^ are considered to a probe band of short-range order of starch. The absorbance at 995 cm^−1^ is associated with the single-helical crystal structure of starch. The bands at 1047 cm^−1^ and 1022 cm^−1^ are related to the short-range ordered structure and amorphous structure of starch, respectively. The ratio of peak intensity at 1047 and 1022 cm^−1^ (R_1047/1022_), which indicates the short-range order of starch, was determined (Table 2). The R_1047/1022_ of dried yam samples treated with HAD, MFD at 0.5 W/g, MFD at 1.0 W/g, and MFD at 1.5 W/g were 0.984, 0.956, 0.963, and 0.969, respectively. A higher R_1047/1022_ value obtained by HAD samples than that of MFD samples was due to the increased mobility of starch chains in the crystalline region, leading to the formation of a double helix [25]. The increased R_1047/1022_ value in MFD-treated samples with increasing microwave power density demonstrated the improvement in short-range order, suggesting that the weaker microwave energy radiation increased the degree of damage to the ordered structure due to the longer drying time. Jiang et al. [23] also suggested that dielectric drying could destroy the structure of starch by de-chaining and hydrolysis.

### 3.4. Micromorphology Observed by SEM

The SEM images of yam flour are shown in Figure 4. The starch granules of Chinese yam are oval in shape and surrounded by a large amount of other substances (e.g., protein, fiber). No significant starch gelatinization was observed in either the HAD or MFD samples. The size of the starch granules varied depending on the drying method and conditions, with the HAD samples having more small-sized granules and the MFD 1.5 W/g sample having more large-sized starch granules. This phenomenon was consistent with the variation of the bulk density of the samples (Section 3.2.1).

### 3.5. Pasting Properties

The pasting profiles of starch offer information on the viscosity and thickening behavior, providing a basis for potential industrial applications. Figure 5 shows the pasting profiles of the yam flours dried by HAD and MFD, and similar trends in viscosity changes are observed. All samples had very low viscosity at the beginning of gelatinization because the starch was not soluble in water at low temperatures. As the temperature of the system increased, the hydrogen bonds in the crystalline region of the starch were broken. Starch granules rapidly absorbed a large amount of water and then swelled, causing the suspension to become a viscous starch gel, accompanied by a gradual increase in viscosity [26]. During the holding stage at a high temperature (95 °C), the viscosity of the dispersion gradually decreased under the action of shearing force. In the subsequent cooling process, the re-association of starch molecules occurred, and a three-dimensional network structure formed, with the released amylose combining with water molecules through hydrogen bonds, so the viscosity increased [12].

The pasting parameters of the dried yam flours prepared by HAD and MFD were compared, as shown in Table 3. The pasting temperature, reflecting the resistance of the starch to swelling and degradation, was not significantly different among the four samples. PV is considered to be the equilibrium point of swelling and rupture of the starch particles during gelatinization, which is often used as an indicator to evaluate the thickening ability [27]. The PV of MFD-treated yam flours at all drying conditions (ranging from 252 to 308 BU) was higher than that of HAD samples (226 BU), indicating that MFD was effective in improving the viscosity of the hot paste and acting as a thickener. This result was associated with better retention of the non-starch components in the yams dried by MFD. It was observed that PV of yam flours dried by MFD at 1.0 W/g was higher than that of samples dried by MFD at 0.5 and 1.5 W/g. This may be related to the disruption of the starch chain structure and the retention of non-starch components due to the combined effect of microwave intensity and drying time. The high PV of the MFD 1.0 W/g samples was associated with low amylose content. Fewer amylopectin fractures occurred in MFD at 1.0 W/g. In addition, a medium level of microwave intensity was better for food nutrient retention than MFD with low microwave intensity but long drying time or high microwave intensity [4]. The retention of non-starch components may contribute to the increase in the PV value.

Similar results to FV were observed in TV. BD is the difference between the peak viscosity and the trough viscosity, reflecting the thermal stability of starch paste. The BD value of yam flour dried by HAD was the smallest, indicating the best stability of hot paste. However, it was worth noting that the high BD value of yam flour dried by MFD partly stemmed from its high PV.

SB is the difference between the final viscosity and the trough viscosity, which reflects the retrogradation behavior of starch. The highest SB value was observed in yam flours treated with MFD at 0.5 W/g and 1.0 W/g, which suggested that more re-association of starch granules occurred. However, the fact that microwave treatment can inhibit the short-term degradation of maize starch during cooling was reported by Yang et al. [28]. The different findings may be attributed to the intensity and duration of the microwaves employed as well as the material properties.

### 3.6. Thermal Properties

Starch gelatinization is the most significant functional attribute in food processing. The effect of microwave power density in MFD on the thermal properties of yam flour is shown in Table 4. The yam flours dried by MFD at 1.0 W/g had the highest *T*_o_ and lowest *T*_p_. High *T*_o_ value indicated the preferential destruction of weak starch crystals. With increasing microwave power density, the increase in *T*_e_ (from 90.08 °C to 92.51 °C) and Δ*H* (from 2.27 to 2.96 J/g) were observed. The higher *T*_p_ and *T*_e_ of yam flours may be due to the more short-range ordered structure of the starch, which was confirmed by FTIR results. However, the fact that high-power microwave treatment caused greater disruption in the crystal structure of maize and potato starch than the low-power microwave was reported by Xu et al. [13]. In contrast, Kumar et al. [29] stated that the increase in microwave treatment duration enhanced the destruction of the crystal structure in potato starch. For MFD, high microwave power density required a short drying time. In this study, the combined effects of microwave intensity and action duration showed that high microwave power and short drying time were conducive to maintaining the ordered structure of the starch.

Due to the enhancement in short-range order of the starch in yam flour with increased microwave power, more heat was required for starch gelatinization. Furthermore, other components such as protein and non-starch polysaccharides in the flour competed with the starch for water [30]. As a result, the increase in Δ*H* of yam flour as microwave power increased was obtained. Compared to the MFD samples, the HAD samples displayed higher *T*_p_, *T*_e_, and Δ*H* values. This was because molecule chains rearranged during HAD in both the crystalline and amorphous regions. As the thermal movement intensified, the internal chemical bond strength and cross-linking of the molecular chains also increased, resulting in a more compact and ordered crystal structure that required more energy to break.

### 3.7. In Vitro Starch Digestibility

In vitro starch digestibility of yam flours subjected to MFD and HAD was evaluated (Table 5). The RDS, SDS, and RS contents of the MFD samples ranged from 14.85% to 20.94%, 47.67% to 57.37%, and 24.63% to 37.48%, respectively. Compared to HAD, MFD increased the RDS and SDS contents and decreased the RS content.

Starch digestibility is largely dependent on its morphological, crystalline, and helical structures. Xu et al. [13] reported that microwave treatment caused the appearance of hollows on the surface of the starch granules, which facilitated digestion by enabling the invasion of enzymes. In contrast, the SEM results of this study showed that the MFD treatment did not result in significant damage to the surface of the starch granules. Because less free water was available during the sublimation drying phase of MFD, the impact of the microwave thermal effect of water molecules on the rupture of starch granules was weaker than when high-moisture samples were treated with microwaves. Therefore, the effect of MFD on starch digestibility may be influenced mainly by the molecular structure.

HAD is similar to an annealing process in that it causes the destruction and rearrangement of amorphous regions, increasing the ordered structure of the starch molecules and thus making the product more resistant to enzymatic hydrolysis [7]. In contrast, MFD samples had the lower RS content and higher SRD and SRD contents. Microwave treatment resulted in the breakage, rearrangement, and degradation of starch molecular chains, which reduced the ordered structure and degree of branching, making them more vulnerable to enzymatic hydrolysis [13,28,31]. As the MFD process involved the differential effects of microwave power, action duration, and even the moisture state and content of the materials, the final result was that samples dried under MFD at 1.5 W/g possessed a higher ordered structure (Table 2) and thus showed higher resistance to enzymatic hydrolysis. MFD at 1.5 W/g took the shortest drying time, and the moisture content when the material underwent phase transition during drying was low. It meant that the high microwave power acted for a short time on yams in which the moisture state was liquid. When the water in the material was in a frozen state, it was difficult for microwaves to be absorbed and acted on starch molecules. While when the phase transition occurred, the presence of liquid water in the material allowed the thermal effect of microwaves generated by dipole polarization to exert. However, the low dielectric constant and loss factor of samples caused by the low moisture content made the thermal effect of microwaves weak. Coupled with the short action time of microwaves, a higher ordered structure in MFD 1.5 W/g treated yams were observed.

## 4. Conclusions

This study showed that, as a freeze-drying technology, MFD took a shorter drying time than HAD, showing outstanding advantages in terms of high efficiency. Meanwhile, MFD gave the product good color and higher water/oil-absorption capacity and solubility for better processing applicability. Among the yam flours treated by MFD, samples dried at 1.5 W/g had the lowest total starch content but the highest amylose content and resistant starch content, which was characterized by the highest gelatinization temperature and enthalpy change. This perhaps resulted from the relative larger granules and the higher degree of short-range order of starch in MFD 1.5 W/g samples. Microwave treatment could cause the breakage, rearrangements, and degradation of starch molecular chains and reduce the short-range ordered structure. Although the microwave intensity was highest at MFD 1.5 W/g, the drying time was the shortest. This study offered a detailed report on the effects of MFD with different microwave intensities on starch-related characteristics of yam flours, and it was speculated that the effects of MFD were comprehensively affected by microwave power, time of action, and even material moisture state and content. It provides a theoretical basis for the production of yam flour for different applications (thickening, resistance to digestion). Considering the different moisture states of the material at various stages of drying in MFD, further research is required to determine the effect of microwave intensity on the quality characteristics associated with the starch of Chinese yam.

## Figures and Tables

**Figure 1 foods-11-02324-f001:**
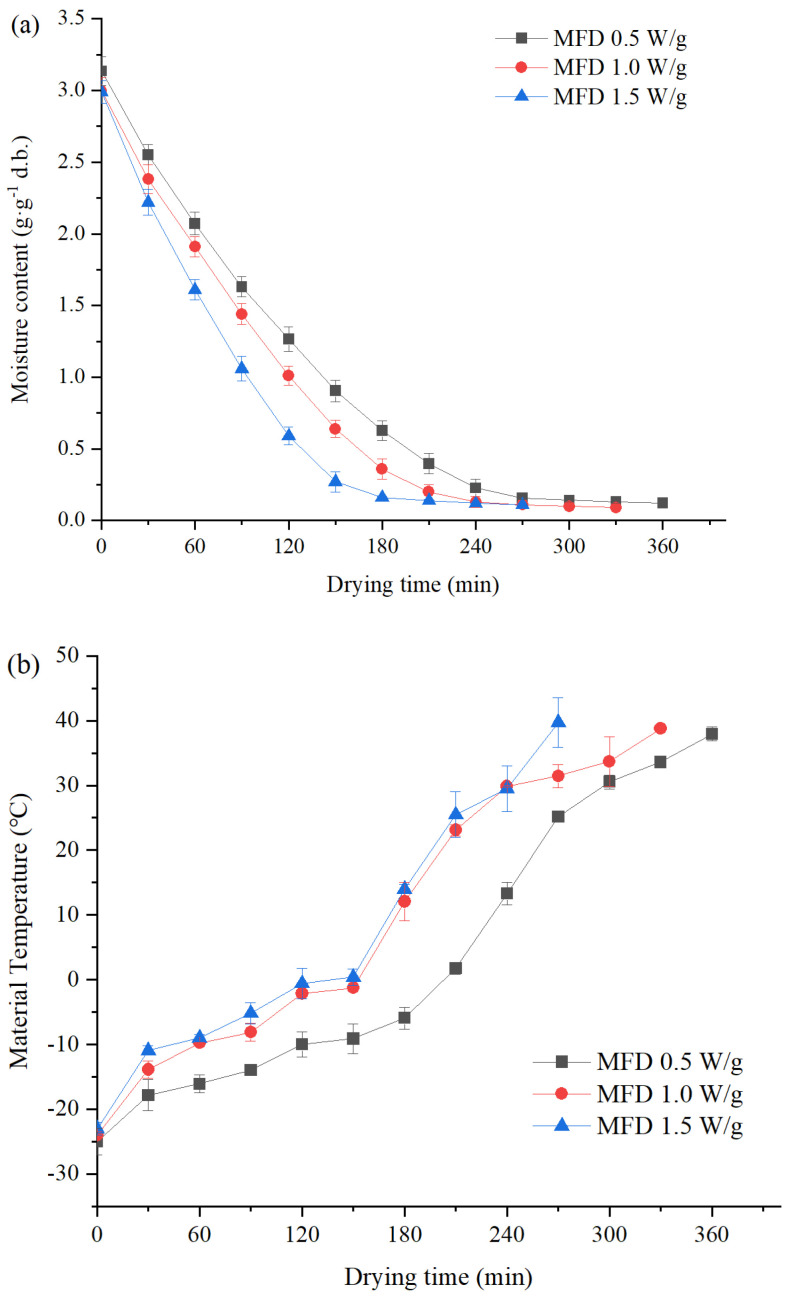
Drying characteristics of MFD for yams under different microwave power densities: (**a**) drying curves and (**b**) temperature profile.

**Figure 2 foods-11-02324-f002:**
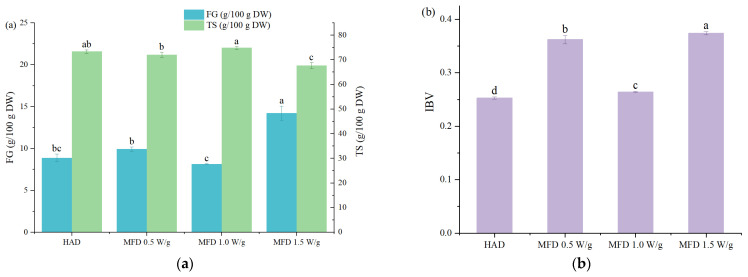
Total starch content, free glucose content (**a**), and iodine blue value (**b**) of yam flours dried by HAD and MFD. Different letters indicate significant differences at *p* < 0.05.

**Figure 3 foods-11-02324-f003:**
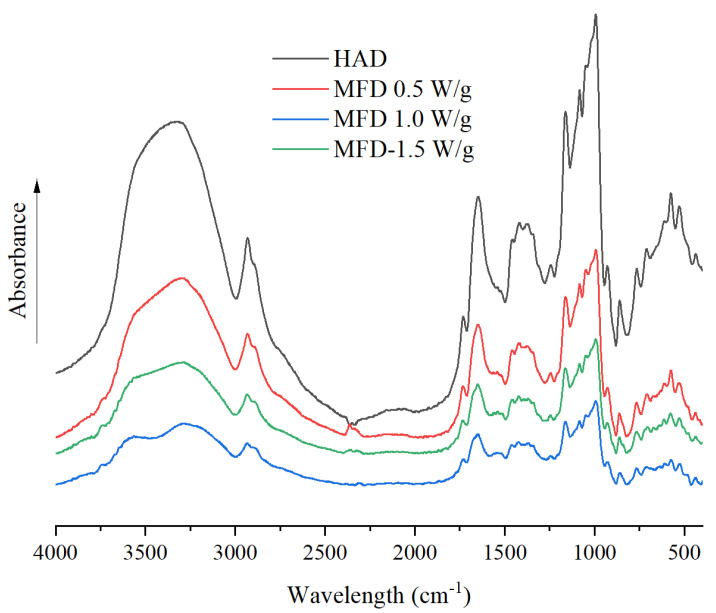
FTIR spectra of yam flours dried by HAD and MFD.

**Figure 4 foods-11-02324-f004:**
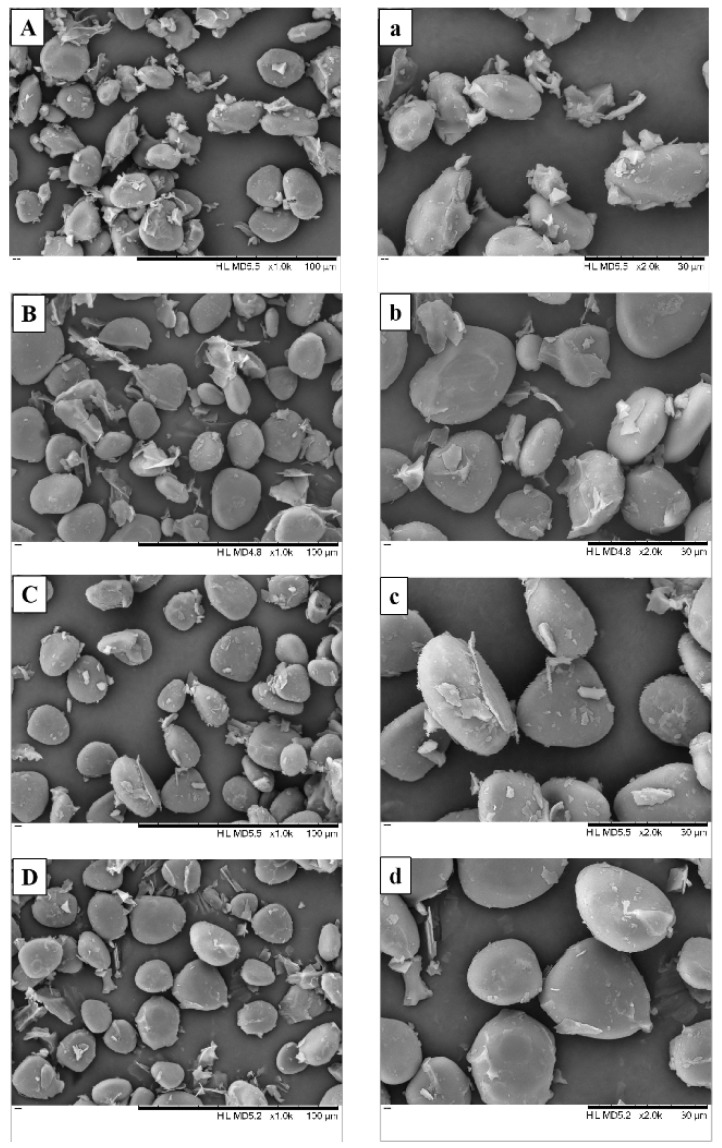
Scanning electron microscopy of yam flours at a magnification of 1000× (**A**–**D**) and 2000× (**a**–**d**). (**A**,**a**) HAD samples; (**B**,**b**) MFD 0.5 W/g samples; (**C**,**c**) MFD 1.0 W/g samples; (**D**,**d**) MFD 1.5 W/g samples.

**Figure 5 foods-11-02324-f005:**
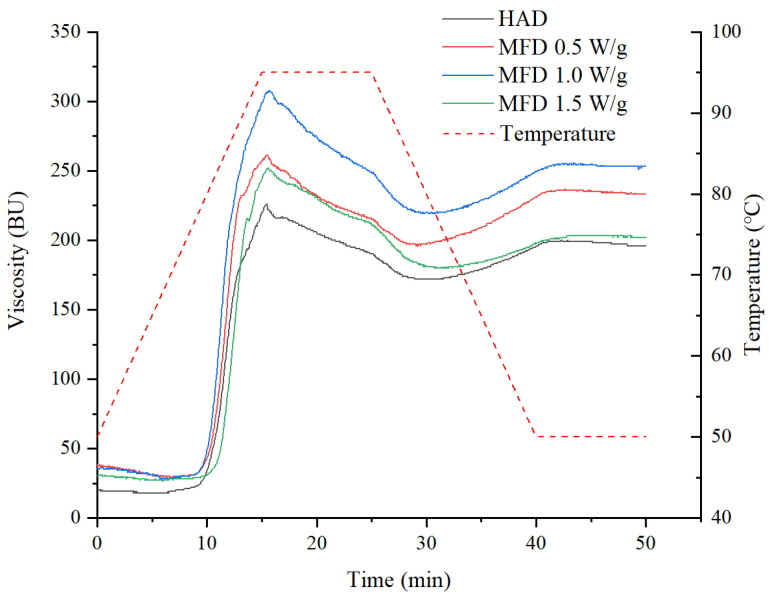
Viscosity–temperature profiles of yam flours prepared by HAD and MFD.

**Table 1 foods-11-02324-t001:** Color and bulk density of dried yam flour processed by HAD and MFD.

Dying Conditions	*L**	*a**	*b**	*WI*	Bulk Density (g/mL)
HAD	91.34 ± 0.22 ^b^	−0.28 ± 0.23 ^a^	8.04 ± 0.69 ^a^	88.16 ± 0.30 ^b^	0.67 ± 0.01 ^a^
MFD, 0.5 W/g	96.49 ± 0.05 ^a^	−1.50 ± 0.08 ^b^	5.14 ± 0.19 ^b^	93.59 ± 0.16 ^a^	0.62 ± 0.04 ^b^
MFD, 1.0 W/g	96.85 ± 0.76 ^a^	−1.67 ± 0.01 ^b^	5.27 ± 0.74 ^b^	93.63 ± 1.00 ^a^	0.58 ± 0.02 ^b^
MFD, 1.5 W/g	96.66 ± 0.08 ^a^	−1.55 ± 0.03 ^b^	4.95 ± 0.05 ^b^	93.83 ± 0.06 ^a^	0.59 ± 0.01 ^b^

Values are expressed as mean ± SD. Different letters in the same column indicate significant differences at *p* < 0.05.

**Table 2 foods-11-02324-t002:** Functional properties and R_1047/1022_ value of dried yam flour processed by HAD and MFD.

Dying Conditions	WAC (g/g)	OAC (g/g)	Solubility (%)	R_1047/1022_
HAD	2.86 ± 0.10 ^b^	0.24 ± 0.07 ^b^	15.81 ± 0.45 ^d^	0.984 ± 0.002 ^a^
MFD, 0.5 W/g	3.31 ± 0.05 ^a^	0.77 ± 0.11 ^a^	19.79 ± 0.83 ^b^	0.956 ± 0.002 ^c^
MFD, 1.0 W/g	3.46 ± 0.03 ^a^	0.80 ± 0.07 ^a^	25.46 ± 0.39 ^a^	0.963 ± 0.005 ^b^
MFD, 1.5 W/g	3.50 ± 0.12 ^a^	0.59 ± 0.06 ^a^	17.44 ± 0.49 ^c^	0.969 ± 0.003 ^b^

Values are expressed as mean ± SD. Different letters in the same column indicate significant difference at *p* < 0.05. WAC, water-absorption capacity; OA, oil-absorption capacity; R_1047/1022_, ratio of peak intensity at 1047 and 1022 cm^−1.^

**Table 3 foods-11-02324-t003:** Pasting properties of yam flours prepared by HAD and MFD.

Drying Conditions	*T*_p_ (°C)	t_p_ (min)	PV (BU)	TV (BU)	FV (BU)	BD (BU)	SB (BU)
HAD	79.0 ± 0.2 ^a^	9 ± 0 ^b^	226 ± 5 ^c^	172 ± 3 ^d^	196 ± 1 ^d^	54 ± 4 ^d^	24 ± 2 ^b^
MFD, 0.5 W/g	79.0 ± 0.4 ^a^	9 ± 0.5 ^b^	261 ± 3 ^b^	199 ± 5 ^b^	233 ± 3 ^b^	62 ± 4 ^b^	34 ± 3 ^a^
MFD, 1.0 W/g	78.7 ± 0.5 ^a^	9 ± 0 ^b^	308 ± 7 ^a^	219 ± 2 ^a^	253 ± 4 ^a^	89 ± 5 ^a^	34 ± 3 ^a^
MFD, 1.5 W/g	79.8 ± 0.7 ^a^	10 ± 0.5 ^a^	252 ± 4 ^b^	181 ± 6 ^c^	202 ± 3 ^c^	71 ± 3 ^c^	21 ± 1 ^b^

Values are expressed as mean ± SD. Different letters in the same column indicate significant difference at *p* < 0.05. *T*_p_, pasting temperature; t_p_, peak time; PV, peak viscosity; TV, trough viscosity; FV, final viscosity; BD, breakdown viscosity; SB, setback viscosity.

**Table 4 foods-11-02324-t004:** Thermal properties of dried yam flours subjected to various drying methods.

Dying Conditions	*T*_o_ (°C)	*T*_p_ (°C)	*T*_e_ (°C)	Δ*H* (J/g)
HAD	74.52 ± 0.13 ^b^	84.78 ± 0.22 ^a^	93.14 ± 1.23 ^a^	3.18 ± 0.12 ^a^
MFD, 0.5 W/g	74.29 ± 0.21 ^b^	83.74 ± 0.21 ^ab^	90.08 ± 0.38 ^b^	2.27 ± 0.21 ^c^
MFD, 1.0 W/g	75.95 ± 0.01 ^a^	83.48 ± 0.63 ^b^	90.40 ± 0.39 ^b^	2.59 ± 0.06 ^bc^
MFD, 1.5 W/g	74.26 ± 0.50 ^b^	84.57 ± 0.24 ^a^	92.51 ± 0.64 ^a^	2.96 ± 0.16 ^ab^

Values are expressed as mean ± SD. Different letters in the same column indicate significant difference at *p* < 0.05. *T*_o_, onset temperature; *T*_p_, peak temperature; *T*_e_, end point temperature; Δ*H*, enthalpy change.

**Table 5 foods-11-02324-t005:** The starch digestibility of yam flours dried by HAD and MFD.

Drying Conditions	RDS (%)	SDS (%)	RS (%)
HAD	10.16 ± 0.01 ^c^	42.88 ± 0.5 ^c^	46.96 ± 0.49 ^a^
MFD, 0.5 W/g	15.99 ± 1.06 ^ab^	57.37 ± 1.18 ^a^	26.64 ± 2.25 ^bc^
MFD, 1.0 W/g	20.94 ± 2.37 ^a^	54.43 ± 2.09 ^ab^	24.63 ± 4.45 ^c^
MFD, 1.5 W/g	14.85 ± 2.45 ^bc^	47.67 ± 4.86 ^bc^	37.48 ± 7.32 ^ab^

Values are expressed as mean ± SD. Different letters in the same column indicate significant difference at *p* < 0.05. RDS, rapidly digestible starch; SDS, slowly digestible starch; RS, resistant starch.

## Data Availability

The data presented in this study are available on request from the corresponding author.

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
