# Peer review of "Physicochemical, Pasting Properties and In Vitro Starch Digestion of Chinese Yam Flours as Affected by Microwave Freeze-Drying"

_foods, 2022, doi:10.3390/foods11152324_

Round 1

Reviewer 1 Report

Article: foods-1834106

Title: Physicochemical, pasting properties and in vitro starch digestion of Chinese yam flours as affected by microwave freeze-drying 

Effects of microwave freeze-drying (MFD) under various microwave power densities (0.5, 1.0, and 1.5 W/g) on the drying characteristics of Chinese yam slices and the physicochemical, pasting, and thermal properties, as well as the starch digestibility of the flour, were investigated using conventional hot air drying (HAD) as a control. The study is a continuity of previous work where different drying technology was studied. The techniques used in the manuscript are important techniques used in the characterization of starch. However, I can not recommend the publication of the manuscript in the present form since many discussions related to starch science are missing, and the format of the manuscript is not present in the form the journal require for publication. For a better understanding, here I present my principal comments.

1. In the literature, there are many techniques for determining amylose, the total amylose by the concanavalin reagent, apparent amylose by the iodine titration, and amylose by gel permeation chromatography; each technique has a different meaning. In this investigation, it seems that the authors determine the leached amylose using the blue value instead of the amylose present inside the granules. When they compare their results with Chen et al 2017, who used amylose content inside granules. Both techniques have different meanings. For this reason and a better correlation with the pasting properties, authors should determine the apparent amylose content in the whole sample and not just the leached amylose.

2. Treatment time (360, 330, and 270 min) and microwave power (0.5, 1.0, and 1.5 W/g) are the main reasons for all the physicochemical changes on the flours obtained. These parameters affected the blue value, but the total starch was not affected at 1.0 W/g. How can you explain this?

3. <Table 1. Fitting results of drying kinetics models for MFD.>

Can you mention what is the porpoise of determining the drying kinetics model? This information seems unimportant for the primary purpose of the manuscript

4. <Explain why did you choose the power 0.5, 1.0, and 1.5 W/g?>

5. increase the magnification of SEM micrograph in order to appreciate in the best way the starch granules

6. <PV of yam flours dried by MFD at 1.0 W/g was higher than that of samples dried by MFD at 0.5 and 1.5 W/g. This may be related to the disruption of the starch chain structure and the retention of non-starch components due to the combined effect of microwave intensity and drying time.>

This argument is contradictory, PV should be higher at 1.5 W/g and not at 1.0 W/g, since, according to FTIR results, the more drying time, the more structural damage on starch granules.

In < 2.8. In vitro starch digestibility properties.>

Could you establish the state of the samples, were they cooked or raw?

In < 3.3.2. Total starch content, free glucose content, and IBV> 

the next sentence is confusing; please double check it

For HAD, a drying temperature of 50°C was appropriate for α-amylase and glucosidase as they had the highest enzyme activity at 55-60°C (Chen et al., 2017), which favored the hydrolysis of starch to glucose. 

Add some reference that supports that microwave treatment can break down starch molecules.

What is WBC?

standardize the terms WOC and OAC

Reviewer 2 Report

Dear  Author,

There are several suggestions for the improvement of the manuscript

1.  Abstract.  Please add more introduction before explaining the aim of the result.  If the result is significant please add information (p <0.05)

2.  Introduction.  Please check the guidelines of the author, and how to cite the reference.  Please also add information regarding the influence of drying on the in vitro digestibility.

3.  Methods.  Please add more information regarding the specification of the instruments used.  There is information regarding the methods for the centrifugation temperature that is not explained.  Please add information about it

3.  Results and Discussion.  Several references are not cited in the manuscript.  Please check

4.  Conclusions.  The conclusion should be aligned with the research's aim and the method generated.  Please provide a short conclusion about the kinetic study performed

Thank you

Round 2

Reviewer 1 Report

Article: foods-1834106

Title: Physicochemical, pasting properties and in vitro starch digestion of Chinese yam flours as affected by microwave freeze-drying 

I must recommend <reconsideration after major revision> based on the following: Because the manuscript does not have individual line numbering in each paragraph, it is difficult to find the authors' changes. In addition, in response to comments on review 1, the accompanying notes given by the authors do not answer all the questions raised. Due to these difficulties, I have to suggest that the authors answer each of the first review questions since I cannot find all your answers at the moment.

Round 3

Reviewer 1 Report

Thank you so much for clarifying the previous comments.